# Preadmission kidney function and risk of acute kidney injury in patients hospitalized with acute pyelonephritis: A Danish population-based cohort study

Henriette Vendelbo Graversen[1]*, Mette Nørgaard[1], Dorothea Nitsch[2], Christian Fynbo Christiansen[1]

1 Department of Clinical Epidemiology, Aarhus University Hospital, Aarhus N, Denmark, 2 Faculty of Epidemiology and Population Health, London School of Hygiene and Tropical Medicine, London, United Kingdom

* au517972@uni.au.dk

## Abstract

### Background and objectives

Only few smaller studies have examined if impaired kidney function increases the risk of acute kidney injury in patients with acute pyelonephritis. Therefore, we estimated 30-day risk of acute kidney injury by preadmission kidney function in patients with acute pyelonephritis. Furthermore, we examined if impaired kidney function was a risk factor for development of acute kidney injury in pyelonephritis patients.

### Methods

This cohort study included patients with a first-time hospitalization with pyelonephritis from 2000 to 2017. Preadmission kidney function (estimated glomerular filtration rate (eGFR) <30, 30–44, 45–59, 60–89, and $\geq$90 ml/min/1.73 m$^2$) and acute kidney injury within 30 days after admission were assessed using laboratory data on serum creatinine. The absolute 30-days risk of acute kidney injury was assessed treating death as a competing risk. The impact of eGFR on the odds of acute kidney injury was compared by odds ratios (ORs) with 95% confidence intervals estimated using logistic regression adjusted for potential confounding factors.

### Results

Among 8,760 patients with available data on preadmission kidney function, 25.8% had a preadmission eGFR <60. The 30-day risk of acute kidney injury was 16% among patients with preadmission eGFR $\geq$90 and increased to 22%, 33%, 42%, and 47% for patients with preadmission eGFR of 60–89, 45–59, 30–44, and <30 respectively. Compared with eGFR$\geq$90, the adjusted ORs for the subgroups with eGFR 60–89, 45–59, 30–45, and <30 were 0.95, 1.32, 1.78, and 2.19 respectively.

**Data Availability Statement:** Data cannot be shared publicly because of Danish legislation. Data can be accessed through the Danish Health Data

Authority for researchers at authorized institutions. Information on data access is available online (http://sundhedsdatastyrelsen.dk/da/forskerservice). Access to data from the Danish Health Data Authority requires approval from the Danish Data Protection Agency (https://www.datatilsynet.dk/english/legislation). The authors did not have special access privileges to these data.

**Funding:** This study was supported by Independent Research Fund Denmark in the form of a grant awarded to CFC (0134-00407B) and the Karen Elise Jensen Foundation in the form of funds awarded to MN. The funders had no role in study design, data collection and analysis, decision to publish, or preparation of the manuscript.

**Competing interests:** The authors have read the journal's policy and have the following competing interests: DN reports grants from GlaxoSmithKline, outside the submitted work. The Department of Clinical Epidemiology is involved in studies with funding from various companies as research grants to (and administered by) Aarhus University. None of these was related to the present study. This does not alter our adherence to PLOS ONE policies on sharing data and materials. There are no patents, products in development or marketed products associated with this research to declare.

## Conclusion

Acute kidney injury is a common complication in patients hospitalized with acute pyelonephritis. Preadmission impaired kidney function is a strong risk factor for development of acute kidney injury in pyelonephritis patients and more attention should be raised in prevention of pyelonephritis in patients with a low kidney function.

## Introduction

Urinary tract infection is a common bacterial infection mainly affecting the lower urinary tract, but the bacteria occasionally ascend and cause upper urinary tract infection, *i.e.* pyelonephritis [1]. The incidence rate of hospitalized pyelonephritis ranges from 1–2 per 10,000 person-years in men to 3–4 per 10,000 person-years in women. The highest incidences are found among infants, young women, and the elderly [2]. Acute kidney injury (AKI) is a potential serious complication to acute pyelonephritis and it is broadly defined by an abrupt decrease in kidney function measured by increasing serum creatinine or lowered urine output [3]. Data on AKI in patients with pyelonephritis are limited. While one in five hospitalized adults may develop AKI [4], a smaller Korean cohort study of 403 patients reported that up to 62.8% of adults hospitalized with acute pyelonephritis developed AKI [5]. Another Taiwanese cohort study of 790 patients reported a 2.63-fold (95% CI 1.53–4.56) increase in the odds of AKI in patients with pyelonephritis compared with patients admitted with lower urinary tract infection [6]. In that study, patients with high age, low baseline estimated glomerular filtration rate (eGFR), and diabetes had the highest odds of AKI [6].

Many previous observational cohort studies examining AKI lacked laboratory information on preadmission serum creatinine [7, 8]. Instead, they identified AKI from diagnostic code, used admission creatinine as baseline value or estimated baseline of creatinine assuming eGFR to be 75 ml/min/1.73m$^2$ [9, 10]. Even though AKI and chronic kidney disease are associated conditions [11–13], only few studies with a small number of patients have examined if impaired kidney function increases the risk of AKI in patients with pyelonephritis [5, 6]. To address the limitation of missing preadmission laboratory information on serum creatinine in the current literature, we took advantage of the nationwide Danish population-based health registries and the potential for identifying preadmission kidney function and AKI from creatinine measurements from laboratory databases. Accordingly, we conducted a large nationwide population-based cohort study to estimate the 30-day risk of AKI by preadmission kidney function. Furthermore, we examined if impaired preadmission kidney function is a risk factor for development of AKI among patients hospitalized with acute pyelonephritis. Such knowledge could direct more attention to prevention of pyelonephritis and associated AKI in those with reduced kidney function to prevent further decline of kidney function [11].

## Materials and methods

### Study design

This historical cohort study conducted in Denmark (population ~ 5.8 million), which has a large amount of routinely collected population-based medical data covering all hospital admissions, hospital diagnoses, medical prescriptions, and any deaths [14]. All Danish citizens have a unique 10-digit civil registration number, which is assigned at birth or immigration, including information on sex and date of birth [15–17]. This number facilitates linkage of

individual-level data between Danish databases including laboratory databases, The Danish Civil Registration System, The Danish National Patient Registry, and The Danish National Prescription Registry [14–16, 18–21]. The Danish health care system is tax-funded and therefore all Danish citizens have free access to medical care at public hospitals. Private hospitals account for less than 1% of all hospital admissions and all acute care, including hospital treatment of pyelonephritis, is provided by public hospitals [14].

Data were accessed through remote access to secure servers at the Danish Health Data Authority after approval by the Danish Health Data Authority (FSEID-00003631) and the Danish Data Protection Agency through registration at Aarhus University (record number 2016-051-000001/812). According to Danish legislation, no ethical approval was required. The data were linked and pseudonymized (i.e., the 10-digit civil registration number was removed) by Danish Health Data Authority before we had data access.

## Study population with pyelonephritis

We included patients 18 or older with a first-time inpatient primary or secondary diagnosis of acute pyelonephritis from 1 January 2000 to 31 December 2017 coded according to the International Classification of Diseases 10th version (ICD-10) recorded in The Danish National Patient Registry [20]. To allow linkage and complete follow-up, we required a valid Danish 10-digit civil registration number including information on sex and date of birth [14–17]. We excluded patients with any ICD-10 diagnosis of chronic tubulo-interstitial nephritis prior to index date (*i.e.* date of hospital admission) to avoid capture of recurrent episodes of pyelonephritis. We additionally excluded patients with missing information on vital status in the 30 days after admission to allow complete follow-up and patients with preadmission chronic dialysis. Patients with muscular dystrophy, paraplegia or tetraplegia were excluded because the muscle atrophy causes low preadmission creatinine and thereby falsely elevated preadmission eGFR, which may bias our findings. In a clinical setting creatinine also would not be used to assess kidney function in these patients [22]. Finally, we excluded patients without any serum creatinine (sCr) measurements within 30 days after hospital admission. These patients are most probably admitted at hospitals not covered by the laboratory databases yet, because we assume that all patients admitted with acute pyelonephritis will have at least one creatinine measurement at admission [23]. Patients lacking outpatient creatinine measurements prior to index date were excluded in the main analyses only including complete cases, but included in the sensitivity analyses. All ICD-10 diagnosis codes used for inclusion and exclusion are listed in S1 Table.

## Variables

**Exposure.** Preadmission eGFR was calculated from the Chronic Kidney Disease Epidemiology Collaboration creatinine equation (CKD-EPI equation) using the most recent outpatient of routine sCr measurement within one year to seven days before the date of pyelonephritis hospitalization [24, 25]. Measurements in the seven days preceding admission were not included to avoid preadmission eGFR being affected of acute illness due to pyelonephritis. Age and sex were used for calculation assuming that all patients were Caucasian, which is a reasonable assumption to make of the Danish population. eGFR was further categorized into <30, 30–44, 45–59, 60–89, and ≥90 corresponding to the eGFR categorization of chronic kidney disease from Kidney Disease Improving Global Outcomes (KDIGO) [26]. Data on sCr were retrieved from the Register of Laboratory Results for Research [21], which contains collected results from inpatients, outpatients, and visitors at the general practitioners from all regions of Denmark except from the Central Denmark region from where we retrieved information from

the regional clinical laboratory information system (LABKA) database [18]. The laboratory databases are increasingly complete throughout the study period, but unfortunately, not all regions of Denmark are covered throughout the study period.

**Outcome.** We followed sCr in all patients from the day at admission and up to 30 days to assess occurrence of AKI *(yes/no)*. According to the guidelines from KDIGO, AKI was defined as an at least 1.5 times relative increase from preadmission sCr, a relative increase of at least 1.5 in sCr within a period of seven days, or an absolute increase in sCr of at least 26.5 μmol/l within a period of 48 hours [3]. Preadmission sCr was defined as the most recent sCr outpatient of routine measurement within one year to seven days before the date of pyelonephritis hospitalization [24, 27]. We did not include urine output in our outcome definition as this information was not available which is generally accepted in AKI-research in a non-intensive-care setting [28].

**Covariates.** Potential confounders were identified through existing knowledge and literature on causal mechanisms and afterwards directed acyclic graphs were constructed [5–7, 29–35]. We therefore collected information on the following potential confounders: age, sex, diabetes, hypertension, malformation of the urinary tract, and heart failure. Information on age and sex were retrieved from the civil registration number through The Danish Civil Registration System [15]. Information on the other variables were found through ICD-8 and ICD-10 diagnosis codes from The Danish National Patient Registry covering all registered diagnosis codes since 1977 [20, 36]. We also used The Danish National Prescription Registry to identify patients treated for diabetes but without any diabetes-related visits at the hospital [19]. We included information on all diagnosis codes since 1977 and all prescriptions since 1994. ICD-8, ICD-10, and ATC codes are listed in Supporting information.

## Statistical analyses

All analyses were performed with Stata software version 14. In Table 1, we described baseline characteristics of patients separated into the five different eGFR categories and patients with missing preadmission baseline creatinine. Sex and comorbidity were described by counts and percentages and age was presented as medians with interquartile ranges.

Main analyses were using only complete cases, *i.e.* excluding all patients with missing information on preadmission kidney function. All analyses were conducted with the patients

**Table 1. Baseline characteristics of the different eGFR categories and the patients with missing preadmission eGFR *(column percentages).***

| Characteristics | eGFR < 30 | eGFR 30–44 | eGFR 45–59 | eGFR 60–89 | eGFR ≥ 90 | Total | Missing preadmission sCr |
|---|---|---|---|---|---|---|---|
| N | 452 (100.00) | 734 (100.0) | 1,076 (100.0) | 3,249 (100.0) | 3,249 (100.0) | 8,760 (100.0) | 7,287 (100.0) |
| **Sex, n (%)** | | | | | | | |
| Male | 213 (47.1) | 331 (45.1) | 445 (41.4) | 1,291 (39.7) | 708 (21.8) | 2,988 (34.1) | 1,414 (19.4) |
| Female | 239 (52.9) | 403 (54.9) | 631 (58.6) | 1,958 (60.3) | 2,541 (78.2) | 5,772 (65.9) | 5,873 (80.6) |
| **Age, median (IQR)** | 79 (70–85) | 79 (71–85) | 76 (68–89) | 69 (59–77) | 43 (29–57) | 65 (46–76) | 38 (25–58) |
| **Comorbidity, n (%)** | | | | | | | |
| Diabetes | 135 (29.9) | 224 (30.5) | 283 (26.3) | 590 (18.2) | 508 (15.6) | 1,740 (19.9) | 380 (5.2) |
| Malformation of the urinary tract | 13 (2.9) | 14 (1.9) | 29 (2.7) | 39 (1.2) | 37 (1.1) | 132 (1.5) | 81 (1.1) |
| Hypertension | 259 (57.3) | 436 (59.4) | 542 (50.4) | 1,165 (35.9) | 490 (15.1) | 2,892 (33.0) | 616 (8.5) |
| Heart failure | 103 (22.8) | 154 (21.0) | 154 (14.3) | 224 (6.9) | 55 (1.7) | 690 (7.9) | 160 (2.2) |
| **Obstructive nephropathy** | 100 (22.1) | 134 (18.3) | 156 (14.5) | 245 (7.5) | 156 (4.8) | 791 (9.0) | - |
| **Nephrolithiasis** | 41 (9.1) | 57 (7.8) | 117 (10.9) | 301 (9.3) | 303 (9.3) | 819 (9.3) | - |
| **Imputed preadmission eGFR, median (IQR)** | - | - | - | - | - | - | 102 (82–119) |

Abbreviations: eGFR, estimated glomerular filtration rate; sCR, serum creatinine; IQR, Interquartile range.

separated into the different exposure groups with a reference group of patients with preadmission eGFR $\geq$90 ml/min/1.73 m$^2$. The patients were included from the date of pyelonephritis hospitalization and followed for 30 days or until an event of AKI or death, whichever came first. We estimated the absolute 30-day risk of AKI and generated a graph of the absolute cumulative 30-day risk of AKI by preadmission kidney function [37]. Death was treated as a competing risk [38]. In the primary analysis, we used logistic regression to determine the association between preadmission eGFR category and AKI within 30 days after pyelonephritis admission. To account for potential confounding, we adjusted for the following covariates: age, sex, hypertension, diabetes, malformation of the urinary tract, and heart failure.

To examine the association between eGFR on a continuous scale without categorization of eGFR and AKI, we modelled a restricted cubic spline function adjusted for the same confounders as the aforementioned logistic regression [39]. The reference value was eGFR 90 ml/min/1.73 m$^2$ and the spline included five knots.

**Sensitivity analyses.** We included and imputed preadmission creatinine for the 7,287 patients with missing preadmission creatinine. We presumed that information on these patients was missing at random because we believe that lacking preadmission creatinine primarily depends on measured variables like age and comorbidity and that is it not directly related to preadmission creatinine level. Imputations were made using information on: age, sex, Charlson Comorbidity Index, congestive heart failure, chronic kidney disease, hypertension, diabetes, cerebrovascular disease, peripheral vascular disease, former recorded usage of catheter a demeure, chronic liver disease, acute dialysis at any time before index, the lowest creatinine measurements in the 30 days after index, and acute dialyses in the 30 days after index [40] (S1 Table). Preadmission creatinine was imputed 50 times and imputed values were averaged for the adjusted logistic regression analysis [40, 41]. We also estimated the absolute 30-day risk of AKI including patients with imputed preadmission values of creatinine. For another sensitivity analysis, we used logistic regression model to examine the association between preadmission eGFR category and AKI within 7 days after admission.

To examine potential effect modification by sex, we conducted a logistic regression analysis stratified by sex and a likelihood ratio test for interaction for the fully adjusted logistic regression with and without an interaction term between age and eGFR. All sensitivity analyses were adjusted for the same covariates as the primary analysis.

We used logistic regression model to conduct a final sensitivity analysis including obstructive nephropathy as a covariate.

## Results

### Patient characteristics

We identified 34,406 adult patients hospitalized with acute pyelonephritis. After applying the first four exclusion criteria, 33,190 patients remained (Fig 1). A total of 16,047 patients of the remaining 33,190 patients were covered by the laboratory databases. Among those patients, 8,760 patients had a creatinine measurement one year to seven days before admission and were included in the primary analysis of complete cases. Patients excluded from the primary analyses because of missing preadmission creatinine were overall younger, more often female, and less comorbid than those included in subsequent analyses (Table 1, last column). Among patients with preadmission creatinine, there were more women than men with pyelonephritis (65.9% vs. 34.1%). The median age was 65 years (IQR 46–76 years). The majority (74.2%) of the patients included in the study had a preadmission eGFR $\geq$ 60 ml/min/1.73 m$^2$, 12.3% had an eGFR between 45 and 59 ml/min/1.73 m$^2$, 8.4% had an eGFR between 30 and 44 ml/min/1.73 m$^2$, and 5.2% had an eGFR below 30 ml/min/1.73 m$^2$ (Table 1). Patients with a lower

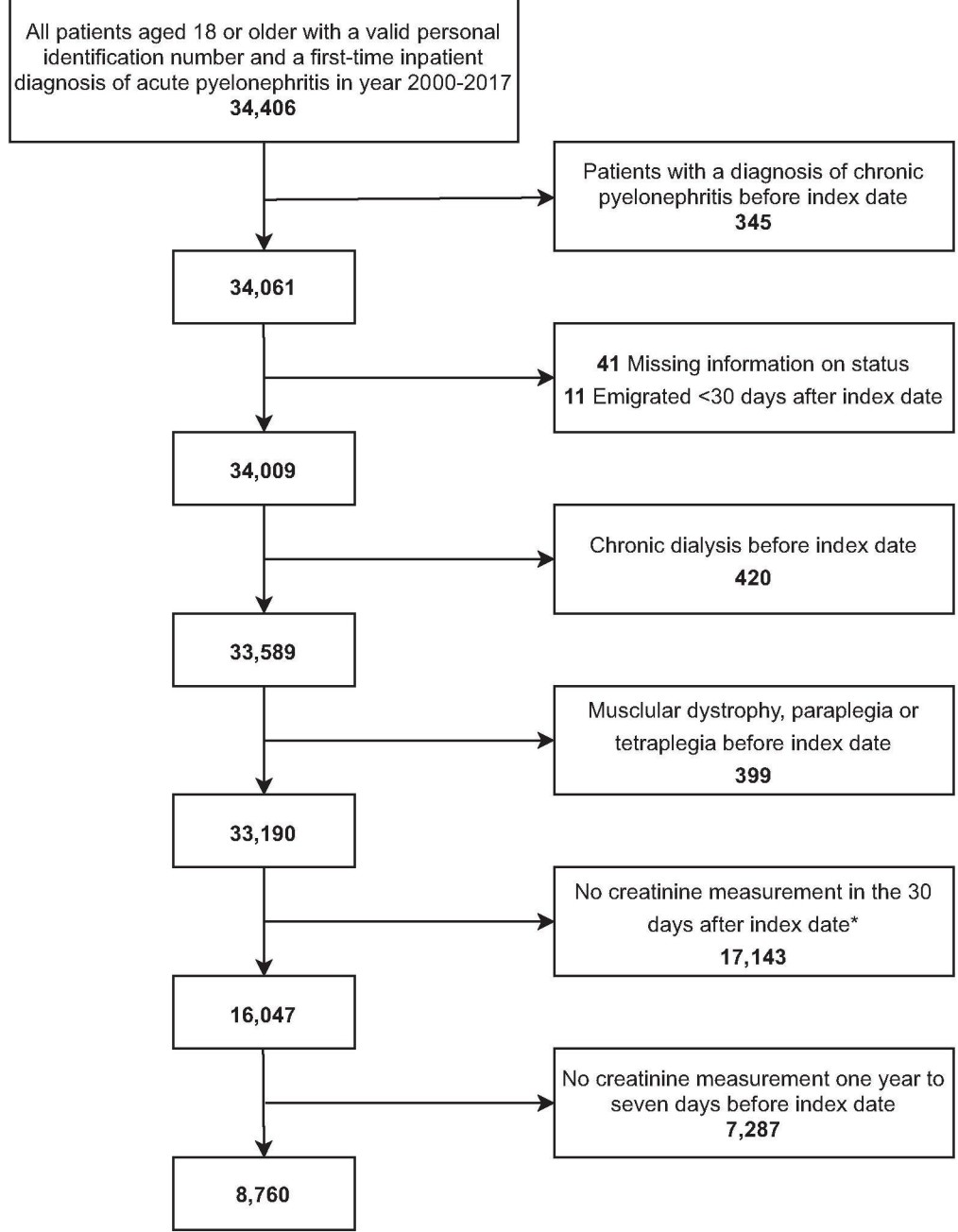

**Fig 1. Flowchart of patient inclusion and exclusions.** *Patients not covered by the laboratory databases.

eGFR tended to be older and include more male patients. In total, 19.9% of all patients had diabetes, 1.5% had a malformation of the urinary tract, 33.0% had hypertension, and 7.9% had a diagnosis of heart failure. The prevalence of comorbidity increased with lower eGFR (Table 1).

## Acute kidney injury

Among the 8,760 patients with a preadmission creatinine, we observed 2,108 incident cases of AKI during the follow-up period. The cumulative 30-day risk of AKI was 24% (95% CI: 23%;

25%) for any cohort participant with pyelonephritis with a total of 64 competing death events. The 30-day risk of AKI increased with lower preadmission eGFR (Fig 2). For patients with an eGFR below 30 ml/min/1.73 m$^2$, the AKI risk was 47% (95% CI: 42%; 51%), which is notable higher than patients with an eGFR $\geq$ 90 ml/min/1.73 m$^2$ who had a 30-day AKI risk of 16% (95% CI: 14%; 17%). This association persisted when adjusting for all included confounders (Fig 3).

In accordance with the ORs from the logistic regression, the ORs from restricted cubic spline model increased with lower eGFR for preadmission eGFR below 90 ml/min/1.73 m$^2$ (Fig 4). The cubic spline was U-shaped and ORs increased in patients with preadmission eGFR above 90 ml/min/1.73 m$^2$.

**Sensitivity analyses.** The findings were confirmed in the sensitivity analysis including imputed preadmission creatinine values, in which we found an overall 30-day AKI risk of 27%. The cumulative 30-day risks of AKI were 24%, 26%, 35%, 44%, and 49% for patients with eGFR $\geq$90, 60–89, 45–59, 30–44 and >30, respectively. In this analysis, we found similar associations between kidney function and AKI, but ORs are slightly higher compared with the primary analysis only including complete cases (S2 Table). A total of 89.3% of the AKI cases (*i.e.* 1,882 cases) occurred in the first week from the day of hospital admission. We found a similar association between kidney function and AKI in the sensitivity analysis of AKI within 7 days after admission, but ORs were slightly lower compared with the primary analysis (S3 Table). In the subgroup analysis stratified by sex, the ORs were comparable with the ORs from the main analysis without firm evidence of interaction (p = 0.07) (S4 Table). The association between preadmission eGFR and AKI attenuated slightly after adjustment for obstructive nephropathy (S5 Table).

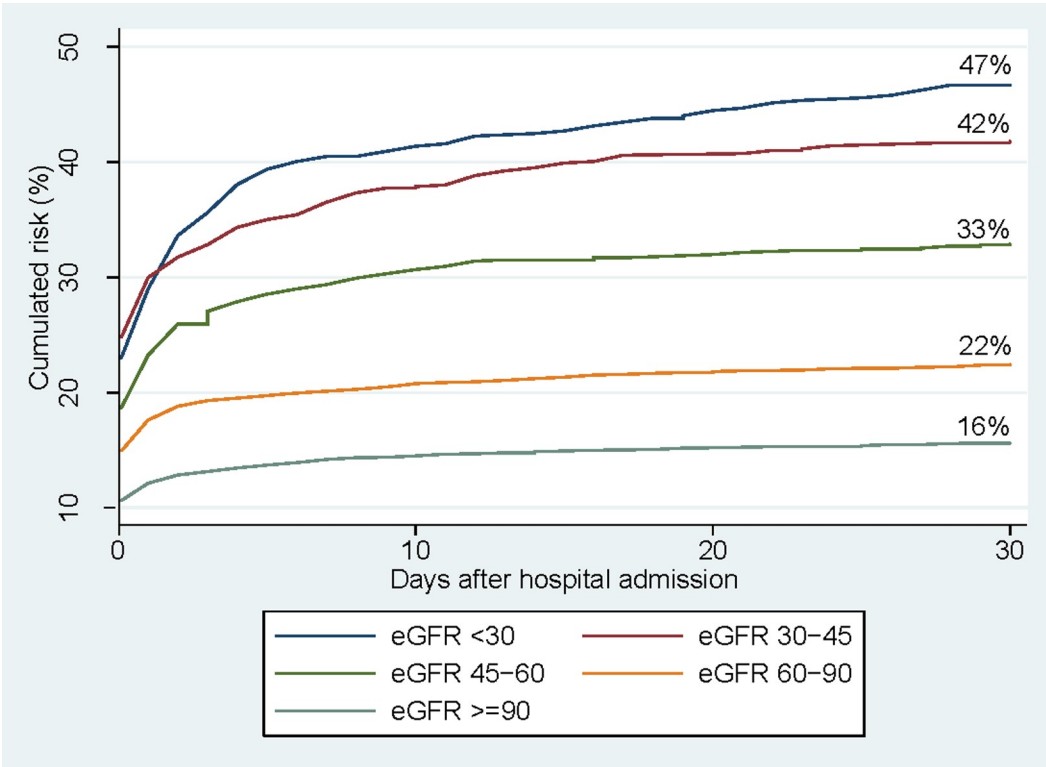

**Fig 2. The cumulative 30-day risk of acute kidney injury after admission with pyelonephritis by different preadmission eGFR categories.** Abbreviations: eGFR, estimated glomerular filtration rate.

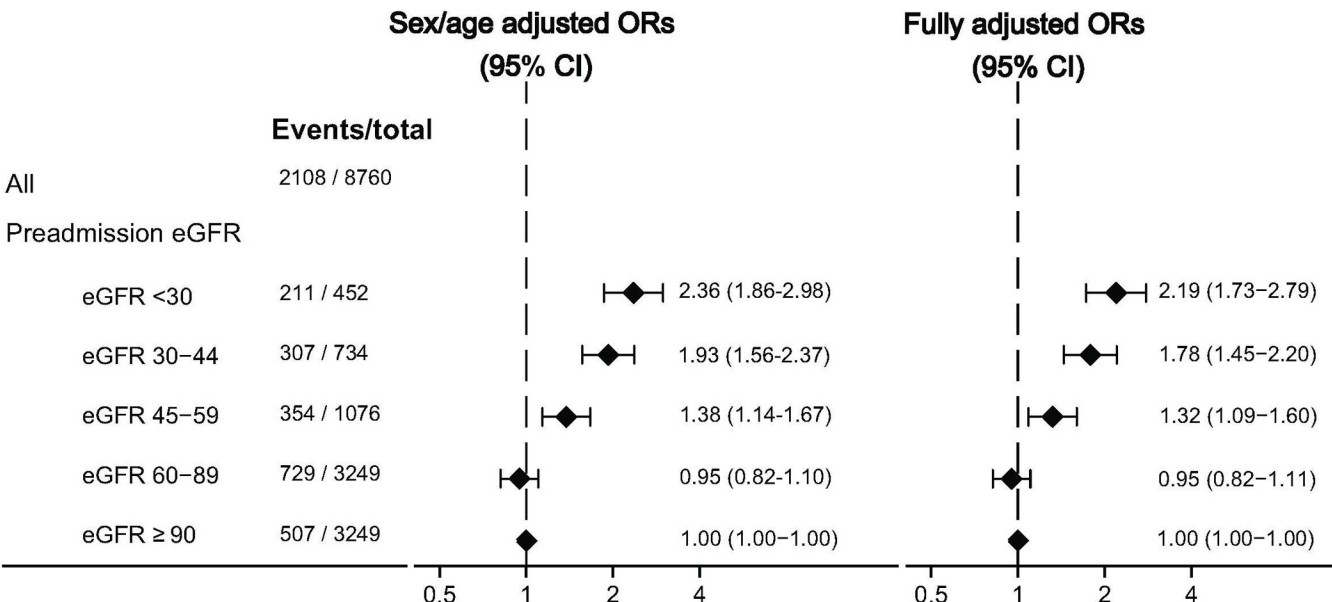

**Fig 3. Forest plot: Events of acute kidney injury within 30 days after pyelonephritis admission/total, and odds ratios from sex/age adjusted and fully adjusted logistic regression analyses by different preadmission eGFR categories.** Abbreviations: ORs, odds ratios; eGFR, estimated glomerular filtration rate. Fully adjusted: adjusted for age, sex, diabetes, hypertension, heart failure, and malformation of the urinary tract.

## Discussion

### Key findings

In this large population-based cohort study including nearly 9,000 patients with a first-time diagnosis of acute pyelonephritis, we found that the overall 30-day risk of AKI was 24% for patients with at least one preadmission creatinine measurement. This 30-day risk of AKI increased in a dose-response relationship with lower preadmission eGFR category and was 47% for patients with eGFR<30ml/min/1.73m$^2$, and was not explained by the included confounding variables.

### Strengths and limitations

This is a large population-based study of pyelonephritis-related AKI, using laboratory information on pre- and post-admission creatinine. Even though the laboratory information was not complete, using laboratory data to identify reduced preadmission eGFR is known to be more accurate than using diagnostic codes, and similarly using creatinine measurements will capture more accurately incident cases of AKI [5, 7, 10, 42–44]. The study has some limitations that should be considered when interpreting the findings. First, we identified our study cohort using the nationwide Danish National Patient Registry including all patients hospitalized and diagnosed with acute pyelonephritis [20]. Still, selection bias may have occurred as we restricted our main analysis to patients with measurements of serum creatinine both before and after admission which means that those included in the complete case analysis are sicker/ older than those excluded [23]. Multiple imputation of missing values did not change our findings, which was reassuring. Second, the regional LABKA database was fairly complete from 2000 and onwards and The Register of Laboratory Results for Research included an increasing number of hospital throughout the study period [18, 21]. However, we do not expect incompleteness to bias our findings because the incompleteness of The Register of Laboratory Results

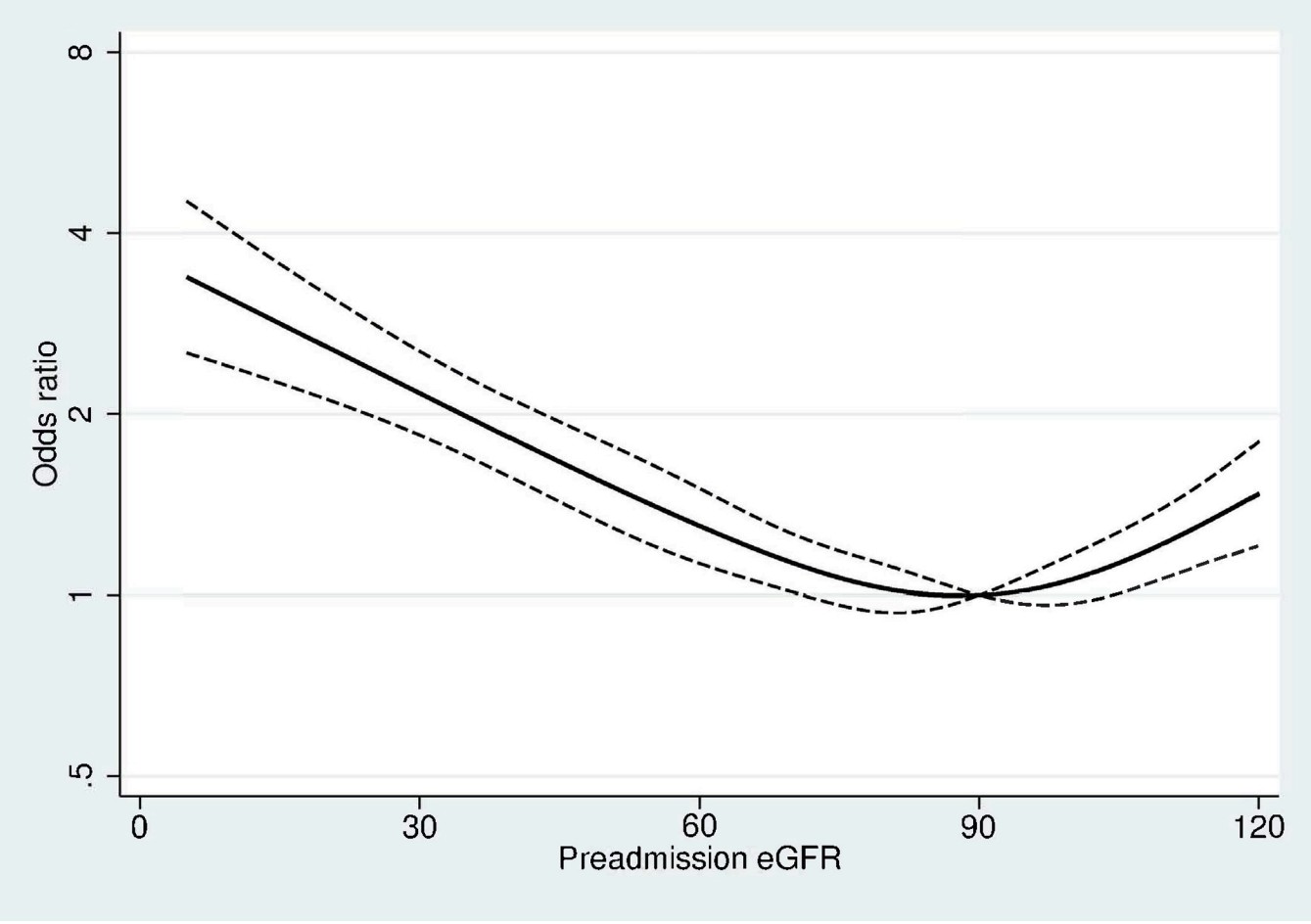

**Fig 4. Graph of the cubic spline model relating exact estimated glomerular filtration rate (eGFR) to the fully adjusted odds ratios (95% CI) of acute kidney injury within 30 days after admission with pyelonephritis.** Abbreviations: eGFR, estimated glomerular filtration rate. Adjusted for age, sex, diabetes, hypertension, heart failure, and malformation of the urinary tract. Preadmission eGFR is calculated most the most recent outpatients creatinine measurement one year to seven days before pyelonephritis admission.

for Research was presumed to be independent of both preadmission kidney function and the risk of AKI. Third, to avoid misclassification of exposure status, we excluded patients without any serum creatinine measurement in the year to seven days prior to admission in the main analyses. These could be patients who were not covered by the laboratory databases, but since patients with missing information on preadmission creatinine were overall much younger and had low comorbidity, it might indicate that most of them were excluded because they simply have not had any creatinine measurements one year prior to admission. Data from the UK suggest that people without any creatinine measurement are very likely to have predominately eGFR>60 ml/min/1.73 m$^2$, so most of these would have contributed to the denominator populations at the higher range of eGFR and reduced the AKI incidence seen for the higher eGFR range–as a result the impact of reduced eGFR on incidence of AKI is probably much more pronounced than estimated by us in this conservative analysis [45]. Fourth, although we included several potential confounders in the analyses, we cannot entirely rule out any unmeasured or residual confounding [29]. Because all our data are from secondary data collection, we lacked information on lifestyle factors, which could have been potential confounders such as body mass index or smoking, though adjustment for comorbid status will have captured the

impact of prolonged poor lifestyle factors. We also lacked information on ethnicity. Approx. 9% of the population have a non-Western background and therefore we cannot exclude any minor bias when assuming that all patients are Caucasian [46]. However, since the majority of the Danish population is Caucasian, this was not expected to influence our results substantially [11, 34].

## Interpretation

Our finding of a 30-day risk of AKI ranging between 16% and 47% depending on preadmission kidney function is consistent with the overall risk of AKI among all hospitalizations found in a meta-analysis by Susantitaphong et al. [4]. However, as is usual in clinical practice all over the world, tests for kidney function are not routinely done in everybody, and data from this meta-analysis are affected by the same limitations as discussed above. Interestingly, we found a U-shaped association between exact eGFR value and the 30-day risk of AKI, but this could in parts be driven by regression to the mean. Results from patients with eGFR$\geq$90 ml/min/1.73 m$^2$ should therefore be interpreted with caution because the CKD-EPI formula is very imprecise at that level of kidney function [26].

Most AKI cases (89.3%) within the 30 days after hospitalization occurred within the first 7 days, which underscores the immediate risk of AKI after hospitalization with pyelonephritis. Complications to pyelonephritis, antibiotic treatment, or underlying disease may contribute to the AKI cases occurring later during the hospitalization.

We found slightly lower ORs when adjusting for obstructive nephropathy (S5 Table). This indicates that some of the found association between preadmission eGFR and AKI could be explained by obstructive nephropathy, but the association persists even after adjustment for this covariate.

Only few smaller studies with a maximum of 790 participants have evaluated the association between impaired kidney function and AKI among patients with pyelonephritis or urinary tract infections [5, 6]. In a smaller study including 403 pyelonephritis patients, Jeon et al. reported an AKI risk as high as 62.8% [5]. We did not find that high risk of AKI, not even among patients with eGFR < 30 ml/min/1.73 m$^2$. This could be due to the fact that they have used single-imputation methods assuming eGFR 75 ml/min/1.73 m$^2$ when preadmission creatinine was missing whereas we have done a complete case analysis. To the best of our knowledge, only one smaller study has evaluated the association between impaired kidney function and AKI with preadmission eGFR categorized into more than two groups [6]. Even though that study also included patients with lower urinary tract infection, they similarly found a dose-response relationship between preadmission eGFR and AKI. Our study thereby extends these previous findings potentially pointing towards a causal association.

The findings of this study suggest that preadmission impaired kidney function is an important risk factor for development of AKI in hospitalized patients with pyelonephritis. Our findings confirm an association between chronic kidney impairment and AKI in patients with pyelonephritis that could not be explained by measured confounding. These findings add one more reason why more attention should be raised in prevention of pyelonephritis associated AKI, perhaps by addressing structural problems, or considering prophylactic treatment or immediate treatment of urinary tract infections with medication stored at home for those with recurrent urinary tract infections, to prevent further worsening of kidney function. This is important because AKI increases the risk of subsequent progression of chronic kidney disease [11]. Therefore, chronic kidney disease patients are a high-risk patient group and more attention should be raised at prevention and treatment of pyelonephritis among these patients.

## Supporting information

**S1 Table. ICD-8, ICD-10, ATC, and procedure codes used for inclusion, exclusion and identification of comorbidity and variables for multiple imputation.** NPU codes and analysis codes for identifying creatinine measurements in the laboratory databases.
(DOCX)

**S2 Table. Odds ratios from logistic regression of acute kidney injury within 30 days after pyelonephritis admission including imputed values on preadmission creatinine from patients missing preadmission eGFR.**
(DOCX)

**S3 Table. Odds ratios from logistic regression of acute kidney injury within 7 days after pyelonephritis admission.**
(DOCX)

**S4 Table. Odds ratios from logistic regression of acute kidney injury within 30 days after pyelonephritis admission stratified by gender including a likelihood ratio test for statistical interaction.**
(DOCX)

**S5 Table. Odds ratios from logistic regression of acute kidney injury within 30 days after pyelonephritis also including obstructive nephropathy as a covariate.**
(DOCX)

**S1 File.**
(DOCX)

## Acknowledgments

We thank Uffe Heidi-Jørgensen and Helene Mathilde Lundsgaard Svane for providing statistical help.

## Author Contributions

**Conceptualization:** Henriette Vendelbo Graversen, Mette Nørgaard, Christian Fynbo Christiansen.

**Data curation:** Henriette Vendelbo Graversen, Mette Nørgaard, Christian Fynbo Christiansen.

**Formal analysis:** Henriette Vendelbo Graversen.

**Funding acquisition:** Christian Fynbo Christiansen.

**Investigation:** Henriette Vendelbo Graversen, Christian Fynbo Christiansen.

**Methodology:** Henriette Vendelbo Graversen, Mette Nørgaard, Dorothea Nitsch, Christian Fynbo Christiansen.

**Project administration:** Henriette Vendelbo Graversen.

**Resources:** Henriette Vendelbo Graversen, Christian Fynbo Christiansen.

**Software:** Henriette Vendelbo Graversen.

**Supervision:** Christian Fynbo Christiansen.

**Validation:** Henriette Vendelbo Graversen.

**Visualization:** Henriette Vendelbo Graversen.

**Writing – original draft:** Henriette Vendelbo Graversen.

**Writing – review & editing:** Henriette Vendelbo Graversen, Mette Nørgaard, Dorothea Nitsch, Christian Fynbo Christiansen.

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
