## [Decision Letter · Decision Letter 0]

11 Dec 2020

PONE-D-20-33791

Preadmission kidney function and risk of acute kidney injury in patients hospitalized with acute pyelonephritis: a Danish population-based cohort study

PLOS ONE

Dear Dr. ,Henriette

Thank you for submitting your manuscript to PLOS ONE. After careful consideration, we feel that it has merit but does not fully meet PLOS ONE’s publication criteria as it currently stands.Therefore, we invite you to submit a revised version of the manuscript that addresses the points raised during the review process.Please  address some of  them comments asked by the  reviewers as the  manuscript may need some minor  revisions before being considered for publication.Please submit your revised manuscript by Jan 23 2021 11:59PM. If you will need more time than this to complete your revisions, please reply to this message or contact the journal office at plosone@plos.org. Please include the following items when submitting your revised manuscript:

We look forward to receiving your revised manuscript.

Kind regards,

Bhagwan Dass, MD

Academic Editor

PLOS ONE

Journal Requirements:

2. Please clarify whether there was any ethical oversight over the study. Was all data used publicly available?

Additional Editor Comments:

1.In your paper ,The criteria used for AKI definition as per the guidelines from KDIGO, AKI was defined as an at least 1.5 times relative increase  from preadmission sCr, a relative increase of at least 1.5 in sCr within a period of seven days, or an absolute  increase in sCr of at least 26.5 µmol/l within a period of 48 hours, I was curious how many cases where  diagnosed based on change in creatnine of at least  26.5 µmol/l and what  percentage were diagnosed based on relative increase in creatnine of at least 1.5 from baseline within a period of seven days .As during episode of  AKI creatnine values tend  to fluctuate before it reaches a steady state.

2.Most of the  time treatment in pyelonephritis varies from 7-14 days depending upon infection underlying condition and organism being treated, just curious to know why the criteria of 30-day risk of acute kidney injury was used. As shorter time frame linked to development of AKI immediately after development of pyelonephritis or to resolution of pyelonephritis makes the correlation easier to undestand  and clinically significant as anything later after resolution of pyelonephritis may add other factors in development of AKI

Reviewers' comments:

Reviewer's Responses to Questions

**Comments to the Author**

1. Is the manuscript technically sound, and do the data support the conclusions?

Reviewer #1: Yes

Reviewer #2: Yes

2. Has the statistical analysis been performed appropriately and rigorously? 

Reviewer #1: Yes

Reviewer #2: Yes

3. Have the authors made all data underlying the findings in their manuscript fully available?

Reviewer #1: Yes

Reviewer #2: Yes

4. Is the manuscript presented in an intelligible fashion and written in standard English?

Reviewer #1: Yes

Reviewer #2: Yes

5. Review Comments to the Author

Reviewer #1: This is a database study that looks at the incidence of AKI in patients admitted with pyelonephritis, stratified by pre-existing CKD. I think that on whole the conclusions support the data. I did notice that the female predominance of pyelonephritis increases with increasing GFR and the male:female ratio at the lowest eGFR was almost equal. That suggests that advanced CKD may predispose to pyelonephritis, which is an interesting observation. I am also interested by the 10% of cases that occurred more than a week after admission. This may be related to antibiotics but probably does not reflect the infection itself. These points should be addressed in the discussion.

Reviewer #2: **Please note that the authors have acknowledged that Danish law prohibits the sharing of health data, and this can only be accessed by researchers who meet specific criteria to access this confidential data.

I would like to thank the authors for their painstaking endeavor to provide an interesting insight into the association of AKI and acute pyelonephritis. I hope to offer some constructive criticism to enhance the scientific objectives of the authors:

Language and stylistic considerations:

Line 116 - "Measurements seven days before admission were not included to avoid preadmission eGFR being affected of acute illness due to pyelonephritis." - For the sake of clarity, please consider changing to ""Measurements in the seven days preceding admission were not included to avoid preadmission eGFR being affected of acute illness due to pyelonephritis."

Line 118 - "...assuming that all patients were Caucasian, which is a reasonable assumption to make of the Danish population." - This is probably a potential source of error for approximately 9% of the population that have a non-Western background (2017 statistics - citation below). I do not disagree with the authors pragmatic approach, but perhaps this potential (relatively minor) source of error could be included as a limitation of the study.

http://www.dst.dk/en/Statistik/emner/befolkning-og-valg/indvandrere-og-efterkommere/indvandrere-og-efterkommere

Line 242-243 - "A total of 89.3% of the AKI cases (i.e. 1,882 cases) 243 occurred within one week after hospital admission." - For the sake of clarity, please consider changing to "A total of 89.3% of the AKI cases (i.e. 1,882 cases) 243 occurred in the first week from the day of hospital admission." The latter sentence makes it clear that the AKI episode occurred relatively early after te acute illness was diagnosed. The original sentence might mislead readers into thinking that AKI may have occurred after the admission concluded (ie. after the patient was discharged).

Methodology / results considerations:

1. Obstructive nephropathy at baseline would be a major risk factor for both chronic kidney disease and for the subsequent development of pyelonephritis. Was this factor considered?

2. Nephrolithiasis, particularly when obstructive, could result in both AKI and pyelonephritis. This would be an important data point to tease out, if possible. Furthermore, even non-obstructive stones can serve as a nidus of infection and result in a higher likelihood of recurrence of infection (and possibly AKI).

In a retrospective analysis, the authors would be limited by the accuracy of ICD-10 data available from these hospitalizations, but I suspect many readers would also want to know if the above points were considered. If this data is not available or too cumbersome to obtain, please consider acknowledging it as potential limitations.

3. Another interpretation which the authors could consider mentioning is that 89.3% of cases of AKI were diagnosed within the first week from the day of admission, which can likely be attributed to sepsis. However, the remaining ~10% of cases do seem somewhat distant from the original septic insult to be directly attributed to acute pyelonephritis. While the authors do not speculate as to the *cause* of the AKI, it is plausible that potentially nephrotoxic agent exposure (IV contrast, certain antibiotics or NSAIDs for pain relief) may account for the delayed AKI cases. It is hard to say if all this needs to be included, but this could make for some thoughtful discussion, as long as the authors feel that it does not digress too far from their topic of research.

6. PLOS authors have the option to publish the peer review history of their article (what does this mean?). If published, this will include your full peer review and any attached files.

Reviewer #1: No

Reviewer #2: **Yes: **Atul Bali, MD

---

## [Author Response · Author response to Decision Letter 0]

1 Feb 2021

Thank you for the carefull review and for the opportunity to revise our papir. We believe the study has improved from the review process and we hope you will find it suitable for publication in PLOS ONE.

On behalf of all the authors,

Henriette Vendelbo Graversen

---

## [Editor Report · Decision Letter 1]

11 Feb 2021

Preadmission kidney function and risk of acute kidney injury in patients hospitalized with acute pyelonephritis: a Danish population-based cohort study

PONE-D-20-33791R1

Dear Dr. Graversen,

We’re pleased to inform you that your manuscript has been judged scientifically suitable for publication and will be formally accepted for publication once it meets all outstanding technical requirements.

Kind regards,

Bhagwan Dass, MD

Academic Editor

PLOS ONE
---

## [Editor Report · Acceptance letter]

17 Feb 2021

PONE-D-20-33791R1 

Preadmission kidney function and risk of acute kidney injury in patients hospitalized with acute pyelonephritis: a Danish population-based cohort study 

Dear Dr. Graversen:

I'm pleased to inform you that your manuscript has been deemed suitable for publication in PLOS ONE. Congratulations! Your manuscript is now with our production department. 

Kind regards, 

on behalf of

Dr. Bhagwan Dass 

Academic Editor

PLOS ONE